# Subtyping *Cryptosporidium xiaoi*, a Common Pathogen in Sheep and Goats

**DOI:** 10.3390/pathogens10070800

**Published:** 2021-06-24

**Authors:** Yingying Fan, Xitong Huang, Sheng Guo, Fang Yang, Xin Yang, Yaqiong Guo, Yaoyu Feng, Lihua Xiao, Na Li

**Affiliations:** 1Center for Emerging and Zoonotic Diseases, College of Veterinary Medicine, South China Agricultural University, Guangzhou 510642, China; fanyingying@webmail.hzau.edu.cn (Y.F.); huangxitong2021@163.com (X.H.); guosheng201911@163.com (S.G.); yf1432277601@163.com (F.Y.); guoyq@scau.edu.cn (Y.G.); yyfeng@scau.edu.cn (Y.F.); 2Guangdong Laboratory for Lingnan Modern Agriculture, Guangzhou 510642, China; 3College of Veterinary Medicine, Northwest A&F University, Yangling, Xianyang 712100, China; xinyang@nwafu.edu.cn

**Keywords:** *Cryptosporidium xiaoi*, 60-kDa glycoprotein, *gp60*, subtyping, genetic diversity, host adaptation

## Abstract

Cryptosporidiosis is a significant cause of diarrhea in sheep and goats. Among the over 40 established species of *Cryptosporidium*, *Cryptosporidium xiaoi* is one of the dominant species infecting ovine and caprine animals. The lack of subtyping tools makes it impossible to examine the transmission of this pathogen. In the present study, we identified and characterized the 60-kDa glycoprotein (*gp60*) gene by sequencing the genome of *C. xiaoi*. The GP60 protein of *C. xiaoi* had a signal peptide, a furin cleavage site of RSRR, a glycosylphosphatidylinositol anchor, and over 100 O-glycosylation sites. Based on the *gp60* sequence, a subtyping tool was developed and used in characterizing *C. xiaoi* in 355 positive samples from sheep and goats in China. A high sequence heterogeneity was observed in the *gp60* gene, with 94 sequence types in 12 subtype families, namely XXIIIa to XXIIIl. Co-infections with multiple subtypes were common in these animals, suggesting that genetic recombination might be responsible for the high diversity within *C. xiaoi*. This was supported by the mosaic sequence patterns among the subtype families. In addition, a potential host adaptation was identified within this species, reflected by the exclusive occurrence of XXIIIa, XXIIIc, XXIIIg, and XXIIIj in goats. This subtyping tool should be useful in studies of the genetic diversity and transmission dynamics of *C. xiaoi*.

## 1. Introduction

*Cryptosporidium* spp. are important diarrheal pathogens in humans and various animals [1]. Currently, 45 *Cryptosporidium* species and over 100 genotypes have been recognized [2]. Among them, *C. parvum*, *C. ubiquitum*, and *C. xiaoi* are common species in sheep and goats. *C. parvum* and *C. ubiquitum* are zoonotic species that infect a wide range of hosts, while *C. xiaoi* appears to be adapted to ovine and caprine animals [3]. *C. xiaoi*, previously known as the *C. bovis*-like genotype, is the most common species in sheep and goats in most areas except Europe [4,5,6,7,8].

Sequence analysis of the 60-kDa glycoprotein (*gp60*) gene has been used extensively in subtyping *C. parvum*, *C. ubiquitum,* and other zoonotic species due to its high sequence heterogeneity and relevance to parasite biology. The unique distribution of subtype families and subtypes have significantly improved our understanding of host adaptation and transmission dynamics within these *Cryptosporidium* spp. [2,9,10,11,12,13,14,15]. Recently, *gp60* gene-based subtyping tools have been developed for molecular epidemiological studies of some non-human pathogenic *Cryptosporidium* spp., such as the bovine-adapted *C. ryanae* and the marsupial-adapted *C. fayeri* [16,17]. However, such a tool is not available for *C. xiaoi*, which is occasionally found in humans [18].

In this study, we sequenced the genome of *C. xiaoi*, identified its *gp60* gene, and developed a subtyping tool for genetic characterizations of isolates from sheep and goats.

## 2. Materials and Methods

### 2.1. Samples

DNA extracts from 434 *C. xiaoi*-positive samples were used in this study, including those from Small Tail Han sheep (*Ovis aries*), Hu sheep (*Ovis aries*), Tibetan sheep (*Ovis aries*), Huanghuai goats (*Capra hircus*), and Black goats (*Capra hircus*) on 11 farms in Qinghai, Henan, Anhui, and Guangdong, China (Table 1). The *C. xiaoi*-positive samples were obtained from previous and ongoing studies of molecular epidemiology of cryptosporidiosis in sheep and goats in China [19]. All the samples were identified as positive for *C. xiaoi* by PCR and sequence analysis of an ~830-bp fragment of the small subunit (*SSU*) rRNA gene [20].

### 2.2. Identification of the gp60 Gene of C. xiaoi

To obtain the nucleotide sequence of the *gp60* gene of *C. xiaoi*, we conducted whole-genome sequencing of one isolate (SCAU2942) from a Hu sheep in Anhui, China using the established procedures [21]. The genome was sequenced using Illumina HiSeq 2500 analysis of an Illumina TruSeq (v3) library with 250-bp paired-end reads. The sequence reads were assembled de novo using the SPAdes version 3.13 (http://cab.spbu.ru/software/spades/, accessed on 21 November 2019) with a K-mer size of 63. The *gp60* gene of *C. xiaoi* was identified by the blastn analysis of the genome assembly with the *gp60* (cgd6_1080) sequence of *C. parvum*. The coding region and amino acid sequence of the *gp60* gene were predicted using the combination of FGENESH (http://www.softberry.com/berry.phtml, accessed on 15 December 2019) and blastp search of the NCBI database.

### 2.3. Subtyping of C. xiaoi

Based on the sequence of the *C. xiaoi gp60* gene, nested PCR primers were designed for the subtyping analysis. The primers used in primary and secondary PCR were Xiaoi-*gp60*-F1 (5′-CCTCTCGGCACTTATTGCCCT-3′) and Xiaoi-*gp60*-R1 (5′-ATACCTGAGATCAAATGCTGATGAA-3′), and Xiaoi-*gp60*-F2 (5′-CCTCTTAGGGGTTCATTGTCTA-3′) and Xiaoi-*gp60*-R2 (5′-TACCTTCAAAGATGACATCAC-3′), respectively. Each PCR was performed in a 50 µL-reaction containing 1×PCR master mix (Thermo Scientific, Waltham, MA, USA), 0.25 µM primary PCR primers or 0.5 µM secondary PCR primers, and 1 µL of DNA (primary PCR) or 2 µL of the primary PCR product (secondary PCR). To reduce PCR inhibitors, 400 ng/µL of nonacetylated bovine serum albumin (Sigma-Aldrich, St. Louis, MO, USA) was used in the primary PCR. The PCR amplification consisted of an initial denaturation at 94 °C for 5 min; 35 cycles of 94 °C (denaturation) for 45 s, 55 °C (annealing) for 45 s, and 72 °C (extension) for 90 s; and a final extension of 72 °C for 10 min. The secondary PCR products were visualized by 1.5% agarose gel electrophoresis.

### 2.4. DNA Sequence Analysis

All secondary *gp60* PCR products were sequenced in both directions using Sanger sequencing by Sangon Biotech (Shanghai, China). For the samples yielding double PCR bands with different sizes, PCR products of each band were excised from the agarose electrophoresis gel and purified using the E.Z.N.A.^®^ Gel Extraction Kit (Omega bio-tek, Norcross, GA, USA) before sequencing. The sequences obtained were assembled using ChromasPro 1.5 (http://technelysium.com.au/wp/chromaspro/, accessed on 20 March 2020), edited using BioEdit 7.1 (http://www. mbio.ncsu.edu/bioedit/bioedit, accessed on 20 March 2020), and aligned with reference sequences from GenBank using MUSCLE in MEGA 7.0 (https://www.megasoftware.net/, accessed on 20 March 2020). Short tandem repeats in the gene were identified using the Tandem Repeat Finder (http://www.tandem.bu.edu/trf/trf, accessed on 21 March 2020). The signal peptide and glycosylphosphatidylinositol (GPI) anchor were predicted using PSORT II (http://psort.hgc.jp/form2.html, accessed on 22 March 2020). N-glycosylated sites, O-glycosylated sites, and furin proteolytic cleavage sites were predicted using NetNGlyc 1.0 (http://www.cbs.dtu.dk/services/NetNGlyc/, accessed on 22 March 2020), YinOYang 1.2 (http://www.cbs.dtu.dk/services/YinOYang/, accessed on 22 March 2020), and ProP 1.0 (http://www.cbs.dtu.dk/services/ProP/, accessed on 22 March 2020), respectively. To assess the genetic relationship of *C. xiaoi* subtype families, a phylogenetic tree was conducted using the maximum likelihood (ML) analysis in MEGA 7.0 based on substitution rates calculated with the general time-reversible model. DnaSP 5.10 (www.ub.es/dnasp/, accessed on 25 March 2020) was used to calculate the recombination rates among subtype families of *C. xiaoi*.

### 2.5. Nucleotide Sequence Accession Numbers

Representative nucleotide sequences of the *C. xiaoi gp60* gene generated in this study were deposited in GenBank under accession numbers MW589389, MW815183-MW815276.

## 3. Results

### 3.1. Features of the gp60 Gene of C. xiaoi

A total of 25.85 million paired-end reads were obtained from the *C. xiaoi* isolate SCAU2942, and assembled into 334,080 contigs. The full *gp60* gene (MW589389) was identified in contig 1122 (8944 bp). The gene was 1437 bp in length and encoded 478 amino acids. Although it shared sequence similarities with the *gp60* gene of *C. parvum* (AF022929), *C. hominis* (FJ839883), *C. ubiquitum* [12], and *C. ryanae* [17] in the 5′ and 3′ regions at the amino acid level, the full sequence similarity was only 19.9 to 41.6% between *C. xiaoi* and the other four species (Figure 1). The GP60 protein of *C. xiaoi* had classic features of *Cryptosporidium* GP60 proteins, including an N-terminal signal peptide, a furin cleavage site (RSRR), two potential N-glycosylation sites, nearly 100 O-glycosylation sites in the GP40 region, and a GPI anchor at the C terminus. Nevertheless, the serine repeats (TCA/TCG/TCT) commonly seen in *C. parvum*, *C. hominis*, and related species, were absent in the 5′ region of the *gp60* gene of *C. xiaoi*.

### 3.2. Sequence Polymorphisms in the gp60 Gene of C. xiaoi

Among the 434 samples positive for *C. xiaoi* in this study, the *gp60* gene in 355 samples (81.8%) was successfully amplified by PCR. PCR products of 323 samples generated one expected band in gel electrophoresis. However, 32 samples yielded two PCR bands with different sizes, including 16 sheep samples and 16 goat samples (Table 1 and Figure 2). All PCR products with either one or two bands were sequenced, generating 298 *gp60* nucleotide sequences with length ranging from 800 to 1170 bp. Nucleotide sequences from 18 samples were identical to the reference sequence (SCAU2942) from the whole-genome sequencing, while the remaining sequences were highly divergent and displayed nucleotide differences of 24.0–68.3% (Table 2). Altogether, 94 sequence types were identified among the 298 *gp60* sequences obtained. In addition to the numerous nucleotide substitutions observed over the partial *gp60* gene, there was a significant length polymorphism among the 94 sequence types mostly due to the presence of repetitive sequences.

### 3.3. Subtype Families and Subtypes of C.xiaoi

A total of 94 *gp60* sequences, including one sequence of each sequence type, were used in a phylogenetic analysis of the *gp60* gene. The ML tree generated comprised 12 clusters of sequences (Figure 3). They were named as subtype families XXIIIa–XXIIIl in concordance with the established nomenclature of *gp60* subtype families of *Cryptosporidium* spp. [22]. Subtype families XXIIIa–XXIIIh formed a group highly divergent from the other group of XXIIIi–XXIIIl (Figure 3). The nucleotide sequence differences among 12 subtype families ranged from 24.0 to 68.3% (Table 2). Among these subtype families, XXIIIl had the shortest nucleotide sequences and contained some unique AGC/AGT trinucleotide repeats encoding serine, leading to the occurrence of a long stretch of highly O-glycosylated amino acids. Subtypes within XXIIIl differed from each other mostly in the number of AGC/AGT trinucleotide repeats. The DnaSP analysis of the *gp60* nucleotide sequences revealed the presence of 71 potential recombination events among all 12 subtype families (Table 2). In addition, mosaic sequence patterns were observed among these subtype families (Figure 4).

At the amino acid level, extensive sequence polymorphism was found among 12 subtype families, mostly in the GP40 region (Figure 4). Despite the extensive sequence difference, all subtype families had the furin cleavage site of RSRR. There were one to four N-glycosylation sites in these subtype families, except for XXIIIk, which had none. In addition, the number of O-glycosylation sites was divergent among subtype families, with XXIIIb and XXIIIl having more O-glycosylation sites than other subtype families.

### 3.4. Distribution of C. xiaoi Subtype Families by Host

Among the 12 subtype families of *C. xiaoi*, XXIIIa (61), XXIIIc (2), XXIIIg (19), and XXIIIj (5) were detected only in goats, while the remaining eight subtype families were found in both sheep and goats. In addition, a common occurrence of co-infections with multiple subtype families was observed in these animals. Among the three breeds of sheep, 64 samples from Han sheep were successfully subtyped, yielding XXIIIb (14), XXIIId (3), XXIIIf (2), XXIIIh (13), XXIIIk (5), XXIIIl (25), XXIIId + XXIIIk (1), and XXIIId + XXIIIl (1); 52 samples from Hu sheep were successfully subtyped, yielding XXIIIb (8), XXIIIe (2), XXIIIf (1), XXIIIh (17), XXIIIk (3), XXIIIl (17), XXIIIh + XXIIIk (2), XXIIIb + XXIIIk, (1) and XXIIId + XXIIIl (1); only a few samples from Tibetan sheep were successfully subtyped, yielding XXIIId (1), XXIIIe (1), XXIIIh (1), and XXIIIi (3). Between the two breeds of goats, only XXIIIa was identified in Black goats, while more divergent subtype families were detected in Huanghuai goats, yielding XXIIIa (26), XXIIIb (7), XXIIIc (2), XXIIId (6), XXIIIe (10), XXIIIf (7), XXIIIg (19), XXIIIh (11), XXIIIi (7), XXIIIj (5), XXIIIk (12), and XXIIIl (22). Noticeably, co-infections of various subtype families were detected in Huanghuai goats, including XXIIIa + XXIIIg (1), XXIIIb + XXIIIl (1), XXIIIh + XXIIIi (1), XXIIIh + XXIIIj (1), XXIIIh + XXIIIk (2), and XXIIIi + XXIIIl (1) (Table 1).

### 3.5. Distribution of C. xiaoi Subtype Families by Farm

One to 10 subtype families were found on each farm. As shown in Table 1, three farms had only one subtype family, two farms had two, two farms had four, one farm had six, one farm had seven, and two farms had 10. On Farms 9, 10, and 11 in Guangdong, all *gp60* sequences obtained belonged to the subtype family XXIIIa. In contrast, although only eight samples were subtyped on Farm 4 in Qinghai, they belonged to four subtype families (XXIIId, XXIIIe, XXIIIh, and XXIIIi). In addition, co-infections of different subtype families were observed in animals on Farms 1, 3, 5, 6, and 7, mostly with prevalent subtype families on the farm (Table 1).

## 4. Discussion

In the present study, we conducted whole genome sequencing of *C. xiaoi* and identified its *gp60* gene. Based on the sequence data, we established a *gp60*-based subtyping tool to assess the genetic diversity of *C. xiaoi*. The application of this subtyping tool in the analysis of *C. xiaoi*-positive samples from various breeds of sheep and goats has identified high genetic diversity within the species and possible differences in the distribution of subtypes between the two types of hosts.

The *gp60* gene sequence of *C. xiaoi* is highly divergent from that of other *Cryptosporidium* spp. Similar to the *gp60* gene of *C. ryanae* (~1548 bp), the *C. xiaoi gp60* gene (~1437 bp) is much longer than those in *C. parvum*, *C. hominis*, and *C. ubiquitum* (~873–1035 bp). Both the nucleotide and amino acid sequences of the *C. xiaoi gp60* gene showed low identity to those of other *Cryptosporidium* spp. This may explain the inability of the commonly used *gp60* primers to amplify DNA of *C. xiaoi* [23]. Similar to *C. ubiquitum*, *C. canis*, *C. felis*, and *C. ryanae*, the trinucleotide repeats of TCA/TCG/TCT encoding a polyserine tract at the 5′ end of the *gp60* gene and widely used to differentiate subtypes within subtype families, were absent in the *gp60* sequence of *C. xiaoi* [12,13,17,24]. However, a polyserine tract encoded by AGC/AGT repeats was observed in the *gp60* gene of the subtype family XXIIIl, and subtypes within XXIIIl differed mostly in the number of AGC/AGT repeats. Similar to most *Cryptosporidium* spp., the GP60 protein of *C. xiaoi* has a classic furin cleavage site “RSRR” between GP40 and GP15, which is absent in *C. ubiquitum*, *C. viatorum*, *Cryptosporidium* chipmunk genotype I and skunk genotype [9,11,12,14].

Based on the sequence analysis, the *gp60* gene of *C. xiaoi* displays an extremely high genetic diversity. The analysis of 298 sequences obtained led to the identification of 94 sequence types in 12 subtype families, including significant length polymorphism and sequence variability. The high sequence heterogeneity in this gene, nevertheless, has made PCR amplification difficult, which together with the large amplicon could be responsible for the poor amplification efficiency. In addition, some samples (13/355) produced double bands in *gp60* PCR, indicating the presence of concurrent infection with different subtypes in sheep and goats. This may facilitate the occurrence of genetic recombination among *C. xiaoi* subtypes, illustrated by the identification of mosaic sequence patterns and 71 potential recombination events in the overall sequence data. Thus, genetic recombination might be responsible for high sequence heterogeneity in the *gp60* gene of *C. xiaoi*. Genetic recombination at the *gp60* locus was observed in *C. parvum*, *C. hominis*, *C. ubiquitum,* and *C. ryanae* [2,12,17].

The *gp60* subtyping results suggest the presence of host adaptation within *C. xiaoi*. Among the 12 subtype families, XXIIIa, XXIIIc, XXIIIg, and XXIIIj were observed only in goats thus far. For the two breeds of goats, Huanghuai goats in Anhui harboured all subtype families of XXIIIa–XXIIIl. In contrast, all 35 samples from Black goats in Guangdong belonged to XXIIIa. The latter could be due to the reduced genetic diversity of *C. xiaoi* in the province. Previously, host-adapted *gp60* subtype families had been identified in other *Cryptosporidium* spp., such as *C. parvum*, *C. hominis*, *C. felis*, *C. ubiquitum*, *C. tyzzeri*, and *C. ryanae* [2,12,17,25,26,27].

No obvious correlation was found between the distribution of *C. xiaoi* subtype families and geographic locations in this study. Even though all *C. xiaoi* isolates from three farms in Guangdong belonged to XXIIIa, this subtype family was found in goats on two farms in Anhui. Subtyping data of *C. xiaoi* from more geographic locations and diverse animals are needed for better understanding of the distribution of *C. xiaoi* subtypes. Previously, geographical differences had been reported in the subtype distribution of *C. hominis*, *C. parvum*, *C. felis*, *C. ubiquitum*, *C. ryanae*, and *Cryptosporidium* chipmunk genotype I, indicating possible differences in the transmission of these pathogens [9,12,17,25,27,28].

## 5. Conclusions

In the present study, we conducted whole-genome sequencing of *C. xiaoi* and developed a subtyping tool based on the *gp60* gene. The application of this new tool in the analysis of fecal samples from sheep and goats has revealed a high genetic diversity within the species, and likely identified the occurrence of host adaptation at the subtype family level. Further studies with extensive sampling of various hosts in diverse areas are needed to improve our understanding of the transmission characteristics of *C. xiaoi*.

## Figures and Tables

**Figure 1 pathogens-10-00800-f001:**
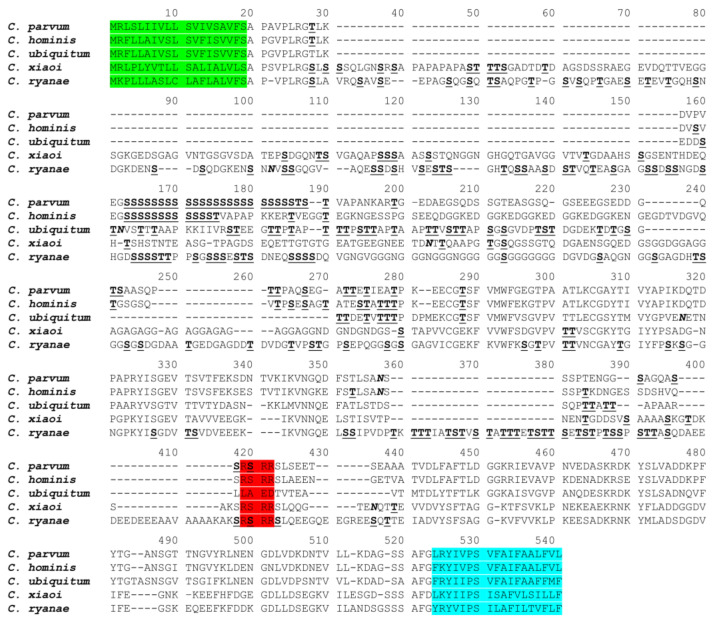
Deduced amino acid sequence of the *gp60* gene of *Cryptosporidium xiaoi* compared with sequences of *C. parvum* (AF022929), *C. hominis* (FJ839883), *C. ubiquitum* [12], and *C. ryanae* [17]. Potential *N*-glycosylation sites are indicated in bold and italic letters, and predicted O-glycosylation sites are indicated in bold and underlined letters. Amino acid sequences of the N-terminal signal peptide, furin cleavage site (RSRR), and C-terminal glycosylphosphatidylinositol anchor are highlighted in green, red, and blue, respectively. Dashes represent amino acid deletions.

**Figure 2 pathogens-10-00800-f002:**
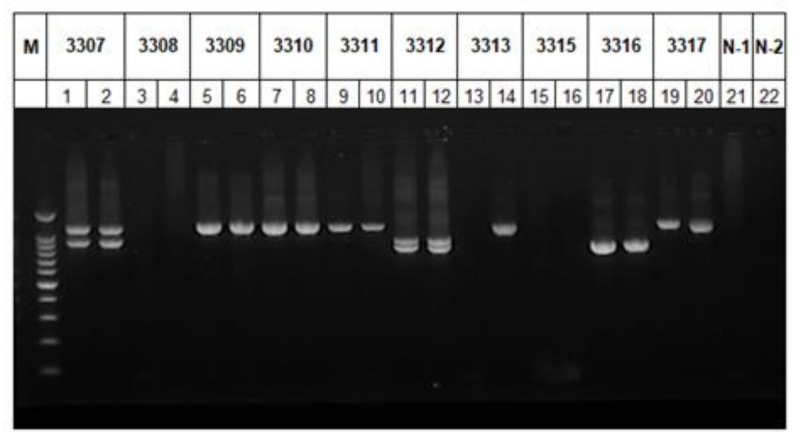
Nested PCR amplification of the partial *gp60* gene of *Cryptosporidium xiaoi* in sheep and goat samples. M: 100-bp DNA ladder. Lanes 1–20: Replicate PCR of 10 samples with divergent binding patterns. N-1 and N-2: No-template controls in the primary and secondary PCR, respectively.

**Figure 3 pathogens-10-00800-f003:**
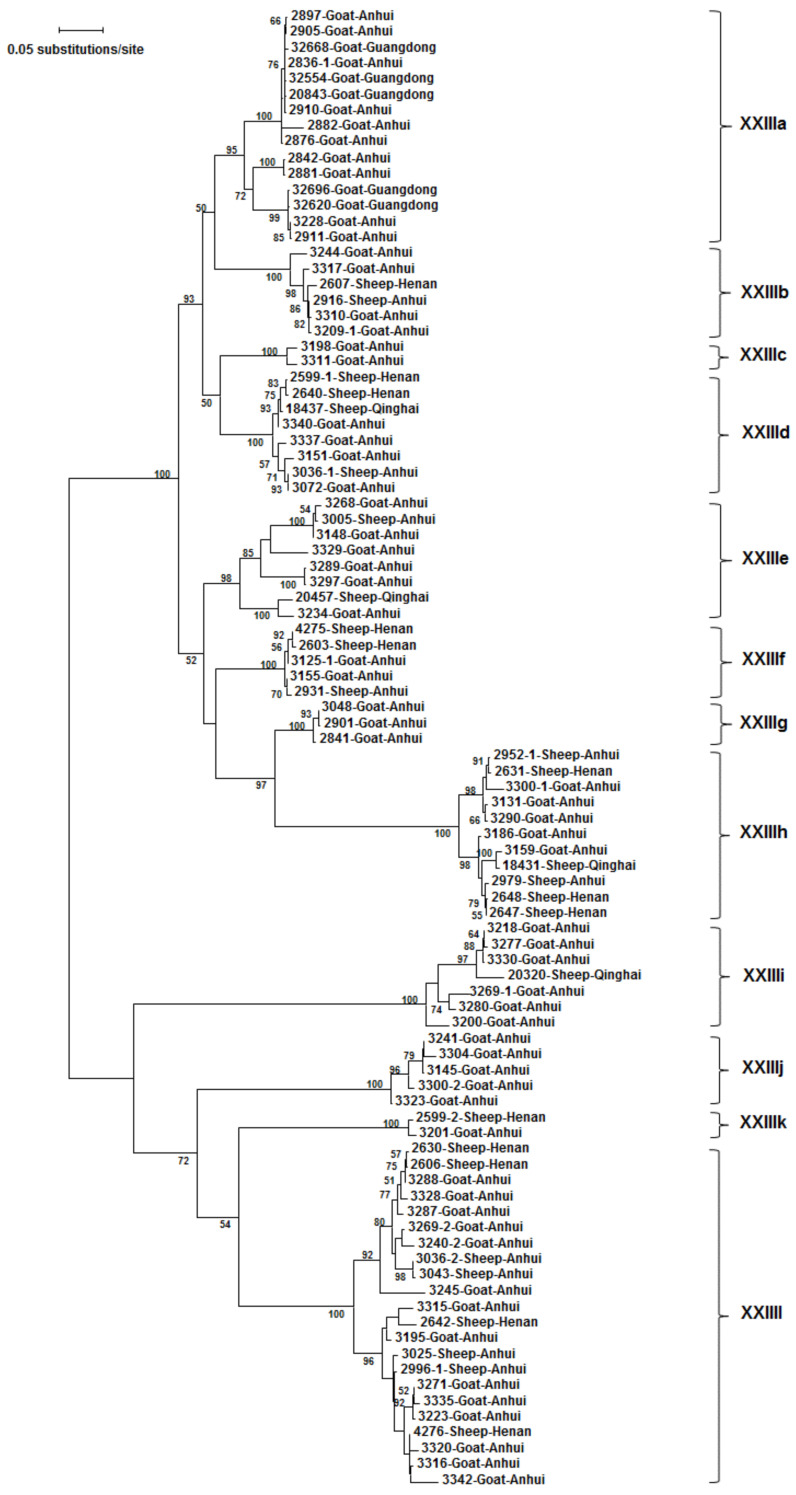
Phylogenetic relationship among 12 *Cryptosporidium xiaoi* subtype families (XXIIIa–XXIIIl) based on the maximum likelihood analysis of the partial *gp60* gene. General time-reversible model and Gamma distribution were used in the calculation of substitution rates. Bootstrap values lower than 50% are not displayed.

**Figure 4 pathogens-10-00800-f004:**
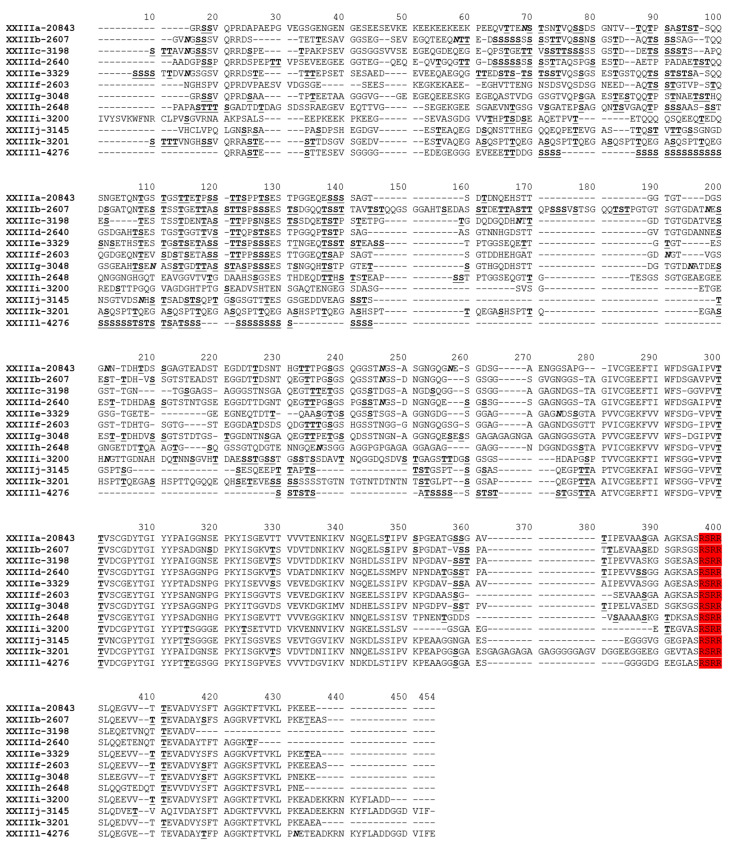
Deduced amino acid sequences of the partial *gp60* gene of 12 subtype families (XXIIIa–XXIIIl) in *Cryptosporidium xiaoi*. N-glycosylation sites are indicated in bold and italic letters, and O-glycosylation sites are indicated in bold and underlined letters. The furin cleavage site “RSRR” is highlighted in red. Dashes represent amino acid deletions (except those at both ends of the sequences).

**Table 1 pathogens-10-00800-t001:** *Cryptosporidium xiaoi* subtype families identified in sheep and goats in China.

Host	Region	Breed	Farm ID	No. of *C. xiaoi*-Positive Samples	No. of Samples Positive at the *gp60* Locus (%)	No. of Samples with Divergent *gp60* PCR Banding Patterns (%)		Subtype Family (no.)
						One Band	Two Bands	
Sheep	Henan	Han	1	71	58 (81.7)	49 (84.5)	9 (15.5)	XXIIIb (13), XXIIId (3), XXIIIf (1), XXIIIh (13), XXIIIl (22), XXIIId + XXIIIk (1), XXIIId + XXIIIl (1)
			2	18	18 (100.0)	18 (100.0)	-	XXIIIb (1), XXIIIf (1), XXIIIk (5), XXIIIl (3)
	Anhui	Hu	3	84	64 (76.2)	57 (89.0)	7 (11.0)	XXIIIb (8), XXIIIe (2), XXIIIf (1), XXIIIh (17), XXIIIk (3), XXIIIl (17), XXIIIh + XXIIIk (2), XXIIIb + XXIIIk (1), XXIIId + XXIIIl (1)
	Qinghai *	Tibetan	4	39	8 (20.5)	8 (100.0)	-	XXIIId (1), XXIIIe (1), XXIIIh (1), XXIIIi (3)
Goats	Anhui	Huanghuai	5	77	77 (100.0)	66 (84.7)	11 (15.3)	XXIIIa (6), XXIIIb (4), XXIIIc (1), XXIIId (2), XXIIIe (5), XXIIIh (7), XXIIIi (4), XXIIIj (4), XXIIIk (5), XXIIIl (18), XXIIIb + XXIIIl (1), XXIIIh + XXIIIi (1), XXIIIh + XXIIIj (1), XXIIIh + XXIIIk (1), XXIIIi + XXIIIl (1)
			6	51	46 (90.2)	43 (93.5)	3 (6.5)	XXIIIb (3), XXIIIc (1), XXIIId (3), XXIIIe (5), XXIIIf (7), XXIIIh (4), XXIIIi (3), XXIIIj (1), XXIIIk (7), XXIIIl (4), XXIIIh +XXIIIk (1)
			7	33	33 (100.0)	31 (93.9)	2 (6.1)	XXIIIa (20), XXIIIg (5), XXIIIa + XXIIIg (1)
			8	20	16 (80.0)	16 (100.0)	-	XXIIId (1), XXIIIg (14)
	Guangdong	Black	9	11	8 (72.7)	8 (100.0)	-	XXIIIa (8)
			10	18	16 (88.9)	16 (100.0)	-	XXIIIa (16)
			11	12	11 (91.7)	11 (100.0)	-	XXIIIa (11)
Total	-	-	-	434	355 (81.8%)	323 (90.1)	32 (9.9)	XXIIIa (61), XXIIIb (29), XXIIIc (2), XXIIId (10), XXIIIe (13), XXIIIf (10), XXIIIg (19), XXIIIh (42), XXIIIi (10), XXIIIj (5), XXIIIk (20), XXIIIl (64), XXIIIa + XXIIIg (1), XXIIIb + XXIIIk (1), XXIIIb + XXIIIl (1), XXIIId + XXIIIk (1), XXIIId + XXIIIl (2), XXIIIh + XXIIIi (1), XXIIIh + XXIIIj (1), XXIIIh + XXIIIk (4), XXIIIi + XXIIIl (1)

* Samples from a previous study [19].

**Table 2 pathogens-10-00800-t002:** Pairwise nucleotide sequence similarity among subtype families of *Cryptosporidium xiaoi* in the *gp60* gene.

	XXIIIa	XXIIIb	XXIIIc	XXIIId	XXIIIe	XXIIIf	XXIIIg	XXIIIh	XXIIIi	XXIIIj	XXIIIk	XXIIIl
**XXIIIa**												
**XXIIIb**	76.0%											
**XXIIIc**	71.2%	74.0%										
**XXIIId**	70.1%	72.7%	68.8%									
**XXIIIe**	71.8%	72.7%	68.3%	71.6%								
**XXIIIf**	74.3%	69.0%	66.3%	70.1%	75.8%							
**XXIIIg**	68.3%	66.2%	66.0%	66.4%	69.8%	71.4%						
**XXIIIh**	54.1%	56.0%	50.4%	55.0%	56.4%	57.1%	57.6%					
**XXIIIi**	41.4%	38.2%	37.9%	42.6%	40.9%	41.4%	39.4%	38.5%				
**XXIIIj**	47.6%	45.3%	43.6%	46.1%	44.8%	45.2%	44.3%	42.4%	47.1%			
**XXIIIk**	37.9%	35.7%	35.5%	37.3%	37.9%	39.4%	37.6%	34.7%	49.8%	50.5%		
**XXIIIl**	33.9%	31.7%	32.2%	35.4%	33.6%	35.3%	33.9%	31.8%	42.3%	42.5%	58.3%	

## Data Availability

Data is contained within the article.

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
