# Peer review of "Subtyping Cryptosporidium xiaoi, a Common Pathogen in Sheep and Goats"

_pathogens, 2021, doi:10.3390/pathogens10070800_

Round 1
Reviewer 1 Report
The manuscript presents the results of the subtyping Cryptosporidium xiaoi, a common pathogen in sheep and goats. In my opinion the article is very interesting and well written.
I have only few minor issues for the authors which should be improved/considered:
- In my opinion would be good if the authors added more information about gp 60 gene in Cryptosporidium sp to the introduction or/and disscusion part.
The authors may cite the manuscript in which this information are contained:
Ikram Rahmouni, Rym Essid, Karim Aoun, and Aïda Bouratbine
„Glycoprotein 60 Diversity in Cryptosporidium parvum Causing Human and Cattle Cryptosporidiosis in the Rural Region of Northern Tunisia".
Am J Trop Med Hyg. 2014 Feb 5; 90(2): 346–350
- The authors should improve/ consider rebuilding the sentence from line 57-58 because it is unclear.
- Futhermore, in my opinion would it be worth while the authors give the manuscript to native speaker or English editing service for improving instances of awkward syntax.
Author Response
The manuscript presents the results of the subtyping Cryptosporidium xiaoi, a common pathogen in sheep and goats. In my opinion the article is very interesting and well written.
RESPONSE: Many thanks for the positive comments.
I have only few minor issues for the authors which should be improved/considered:
- In my opinion would be good if the authors added more information about gp60 gene in Cryptosporidium spp. to the introduction or/and discussion part. The authors may cite the manuscript in which this information are contained: Ikram Rahmouni, Rym Essid, Karim Aoun, and Aïda Bouratbine, Glycoprotein 60 Diversity in Cryptosporidium parvum Causing Human and Cattle Cryptosporidiosis in the Rural Region of Northern Tunisia. Am J Trop Med Hyg. 2014 Feb 5; 90(2): 346–350.
RESPONSE: Thanks for the suggestion. We have added the following information in Lines 42-47 and cited this reference. “Sequence analysis of the 60-kDa glycoprotein (gp60) gene has been used extensively in subtyping C. parvum, C. ubiquitum and other zoonotic species because of its high sequence heterogeneity and relevance to parasite biology. The unique distribution of subtype families and subtypes have significantly improved our understanding of host adaptation and transmission dynamics within these Cryptosporidium spp.”
- The authors should improve/ consider rebuilding the sentence from line 57-58 because it is unclear.
RESPONSE: Thanks for the suggestion. We have modified the sentence as “The C. xiaoi-positive samples were obtained from previous and ongoing studies of molecular epidemiology of cryptosporidiosis in sheep and goats in China” in Lines 61-63.
- Furthermore, in my opinion would it be worth while the authors give the manuscript to native speaker or English editing service for improving instances of awkward syntax.
RESPONSE: Thanks for the suggestion. We have the manuscript edited by a co-author who worked in the United States for 32 years. Please see the tracked version of the revised manuscript for details.
Reviewer 2 Report
This very interesting study describe a subtyping method of Cryptosporidium xiaoi based on the gp60gene.
This method was used to analyse i fecal samples from sheep and goats and revealed high genetic diversity within C. xiaoi, with dsomes subtypes present only in certain hosts.
The paper is clear, and the method described will be very useful for the characterization of this parasite species.
Author Response
This very interesting study describe a subtyping method of Cryptosporidium xiaoi based on the gp60 gene. This method was used to analyze fecal samples from sheep and goats and revealed high genetic diversity within C. xiaoi, with some subtypes present only in certain hosts. The paper is clear, and the method described will be very useful for the characterization of this parasite species.
RESPONSE: Many thanks for the positive comments.